# Identification of Prognostic Markers of Gynecologic Cancers Utilizing Patient-Derived Xenograft Mouse Models

**DOI:** 10.3390/cancers14030829

**Published:** 2022-02-06

**Authors:** Ha-Yeon Shin, Eun-ju Lee, Wookyeom Yang, Hyo Sun Kim, Dawn Chung, Hanbyoul Cho, Jae-Hoon Kim

**Affiliations:** 1Department of Obstetrics and Gynecology, Gangnam Severance Hospital, Yonsei University College of Medicine, Seoul 06273, Korea; hayeon37@yuhs.ac (H.-Y.S.); forgetme107@gmail.com (E.-j.L.); khs88@yuhs.ac (H.S.K.); dawny1004@yuhs.ac (D.C.); hanbyoul@yuhs.ac (H.C.); 2Severance Biomedical Science Institute, Yonsei University College of Medicine, Seoul 03722, Korea; wookyeom@yuhs.ac

**Keywords:** patient-derived xenograft, gynecologic cancer, prognostic markers

## Abstract

**Simple Summary:**

As a preclinical model for personalized cancer therapy, the patient-derived xenograft (PDX) model is suitable because it provides the best representation of the original tumor, but it still has many limitations. We analyzed gynecologic cancer PDX models with their clinical information and gene expression profiling, and found that the success rate of PDX correlated with the patient’s tumor grade and prognosis. Moreover, we showed that the faster the tumor progressed in PDXs, the poorer the prognosis in ovarian cancer patients. We confirmed that the differentially expressed genes (DEGs) selected according to PDX engraftment status could be a prognosis marker of ovarian clear cell cancer patients. This study paves the way for a better application of the PDX in gynecologic cancer.

**Abstract:**

Patient-derived xenografts (PDXs) are important in vivo models for the development of precision medicine. However, challenges exist regarding genetic alterations and relapse after primary treatment. Thus, PDX models are required as a new approach for preclinical and clinical studies. We established PDX models of gynecologic cancers and analyzed their clinical information. We subcutaneously transplanted 207 tumor tissues from patients with gynecologic cancer into nude mice from 2014 to 2019. The successful engraftment rate of ovarian, cervical, and uterine cancer was 47%, 64%, and 56%, respectively. The subsequent passages (P2 and P3) showed higher success and faster growth rates than the first passage (P1). Using gynecologic cancer PDX models, the tumor grade is a common clinical factor affecting PDX establishment. We found that the PDX success rate correlated with the patient’s prognosis, and also that ovarian cancer patients with a poor prognosis had a faster PDX growth rate (*p* < 0.0001). Next, the gene sets associated with inflammation and immune responses were shown in high-ranking successful PDX engraftment through gene set enrichment analysis and RNA sequencing. Up-regulated genes in successful engraftment were found to correlate with ovarian clear cell cancer patient outcomes via Gene Expression Omnibus dataset analysis.

## 1. Introduction

Gynecologic cancers are defined as cancers occurring in the human female reproductive system, composed of internal and external sex organs [1]. Approximately 113,520 American women are diagnosed with gynecologic cancer each year, and 33,620 die every year as result of this cancer [2]. In Korea, the incidence rates for gynecologic cancers increased annually from 2005 until 2017. The 5-year relative survival rates from ovarian, cervical, and endometrial cancer were reported to be 61%, 81%, and 88%, respectively [3]. Gynecologic cancers frequently occur in postmenopausal women, with a high incidence rate in women aged 50–60 years [4,5]. However, in recent years, the incidence of gynecologic cancer has increased in women under the age of 39 and over the age of 80 [6]. Gynecologic cancer does not only occur in postmenopausal women; it can occur in women of all ages. The emergence of this cancer in a period of active living places an enormous burden on patients’ quality of life. Therefore, gynecologic cancer has become a major burden throughout women’s lives.

Comprehensive therapy for gynecologic cancer includes surgery, platinum/paclitaxel-based chemotherapy, and radiotherapy with intensity-modulated radiation therapy (IMRT) [4]. Recently, the application of various targeted therapies and immune checkpoint inhibitors has improved clinical outcomes [7,8,9]. However, there is still a need to improve therapy options for patients with advanced-stage disease. Theoretically, cervical cancers can be overcome through vaccination and regular screening, but many patients still miss the appropriate treatment time [10]. A new milestone in ovarian cancer therapy is thought to be the use of vascular endothelial growth factor inhibitors and poly (ADP-ribose) polymerase inhibitors in patients with homologous recombination deficiency [11]. However, evidence for biomarker-driven treatment is lacking. Moreover, as less than half of patients have non-homologous recombination deficiency, new drug therapies are required. Most endometrial cancers are treated successfully with surgical debulking alone [12]. Endometrial cancers can predict the prognosis of endometrioid adenocarcinoma through molecular classification, which can help determine adjuvant treatment [13,14]. However, the development of effective therapeutic agents for a small number of patients with advanced or recurrent disease is required. Therefore, it is necessary to determine the most effective and appropriate drugs for each patient.

Patient-derived xenografts (PDXs) are used to directly implant fresh tumor tissues from patients with cancer into immune-deficient mice. The advantages of PDX models are well-maintained tumor heterogeneity and tumor microenvironment, including cellular complexity, genomic, stromal architecture, and non-tumor cells [15,16,17]. Hence, the therapeutic effect of drugs is better evaluated by PDX models than by other in vivo models such as cell line xenografts, syngeneic models, and genetically engineered mouse models [18]. By applying PDX models to various cancers, we have observed that the tumor engraftment rate differs according to tumor type. PDXs have a high success rate for colorectal cancer (76–89%) and head and neck cancer (68–85%), but a low success rate for breast cancer (13–21%) and liver cancer (14%) [19]. The success rates of gynecologic cancers have been reported as 18.5–74%, 48–70%, and 60% in ovarian [20,21,22], cervical [23,24], and endometrial cancer [25], respectively. Moreover, the success rates have not been reported for rare gynecologic cancers, such as vaginal and vulvar cancers [26].

Initially, PDX models were designed to screen anti-cancer drugs for personalized medicine. This is because PDX tumor tissues have similar clinicopathological characteristics for each patient [27,28,29]. However, such efforts were soon challenged because it was uncertain whether the PDX establishment was successful, and a substantial period of time was also required before drug testing [30]. We often encountered cases in which the disease progressed before the patient’s PDX model was completed, and the treatment time was missed, or the patient’s disease relapsed and had different characteristics, making PDX models ineffective. Henceforth, we need a comprehensive database of PDX characteristics and the efficacy of various anticancer drugs on PDX. The reason is that it is possible to quickly identify a drug with good effect in the PDX model most similar to the patient’s tumor via this database when a patient is diagnosed with cancer.

According to previous reports, various factors influence the establishment of PDX, but more aggressive and poor prognostic types show a higher PDX success rate. These phenomena tend to be consistent regardless of tumor origin [21,31,32]. Based on these phenomena, the prognosis of patients can be predicted by comparing gene expression alterations between cancer tissues in PDX engraftment success and failure. This study aimed to establish various gynecological PDX models and identify clinicopathological factors influencing PDX establishment from patient clinical information. Additionally, we evaluated whether the differentially expressed genes (DEGs) according to PDX engraftment status could predict patient prognosis.

## 2. Materials and Methods

### 2.1. Patients and Tissue Specimens

This study was approved by the Institutional Review Board of Gangnam Severance Hospital (3-2014-0184; Seoul, Korea). A total of 207 patients with gynecologic cancer from 2014 to 2019 were enrolled in this study. The experiments were performed with each patient’s understanding and written consent, following the Declaration of Helsinki. After tumor surgery, the specimens were immediately sectioned. To establish PDX models, a portion was transferred to the laboratory under transport media with 1% penicillin/streptomycin, 5 ug/mL tetracycline, and 10 ug/mL ciprofloxacin. A portion was quickly submitted for formalin-fixed paraffin embedding, and other portions were stocked with preservation solution, RNAlater^TM^ (Invitrogen, Carlsbad, USA), or without RNAlater^TM^ at −150 °C or −80 °C.

### 2.2. Engraftment and Management of PDX Models

For engraftment in mice, the animal experiments were approved by the Institutional Animal Care and Use Committee at Yonsei University (2014-0273, 2018-0047; Seoul, Korea). The tumors sectioned to approximately 3 mm^2^ were subcutaneously transplanted into at least two female 5- to 6-week-old BALB/c nude mice (four subcutaneous transplants per mouse). First, the mice were anesthetized by injection of a 100 µL mixture of Zoletil/Rompun/DDW (1:1:7 ratio). The surgery site on each flank of the nude mice was disinfected with povidone-iodine pads and 70% isopropyl alcohol swabs. The site was incised, a piece of tissue was implanted, and finally, the site was sutured using a stitching fiber. The tumor size was evaluated using a digital caliper. The tumor volume was calculated using the formula: Volume = (Width^2^ × length) × 0.5. We observed tumor growth for one year. When the size of the biggest tumor among the tumors implanted and growing in the mice was over 100 mm^3^, we defined it as successful engraftment. The xenograft tumors engrafted into mice from patient-derived tumors were defined as P1. After that, xenograft tumors serially engrafted into the next mice were termed as P2 to P3. We collected and banked small pieces of tumor tissue from each passage of PDX tissue.

### 2.3. Tumor Growth Rate

We calculated tumor growth rate because the time required for growth and tumor volume differs for each PDX tumor.
(1)$$\mathrm{Tumor}\;\mathrm{growth}\;\mathrm{rate}=\frac{\mathrm{Final}\;\mathrm{tumor}\;\mathrm{volume}\;({\mathrm{mm}}^{3})}{\mathrm{Duration}\;\left(\mathrm{month}\right)}$$

Final tumor volumes are to be measured just before mice are sacrificed for the next passage. Here, duration implies a period of development of PDX from tissue transplantation to the next passage.

### 2.4. Gene Set Enrichment Analysis (GSEA) and Gene Clustering

GSEA was performed to identify DEGs that were enriched in the gene lists extracted from MSigDB3 [33].

### 2.5. Histologic and Immunohistochemical Analysis

Tissues were fixed overnight in 10% formalin and processed in the tissue core facility at the Mayo Clinic. To compare with the histopathologic features of tumor tissue, both human and mouse tumor tissues were divided into 10 um sections and stained with hematoxylin and eosin (H & E). For immunohistochemistry, deparaffinized and rehydrated 10 um sections were retrieved in pH 6.0, or 9.0 citrate buffer (Agilent Technologies, Inc., Santa Clara, CA, USA). Then, the sections were inactivated of endogenous peroxidase with 3% hydrogen peroxide (Duksan, Seoul, Korea) and incubated with 5% BSA. Next, the cells were incubated with the primary antibody for 1 h at room temperature. The secondary antibody was applied for 30 min at room temperature, followed by detection using 3,3′-diaminobenzidine chromogen solution (Agilent Technologies, Inc.). The sections were mounted.

### 2.6. Antibodies

The primary antibodies used for immunohistochemistry were as follows. The anti-NFKB2 antibody (#15503-1-AP, anti-RELB antibody (#10544), and anti-ICOSLG antibody (TA808779) were purchased from Proteintech Group (Rosemont, IL, USA), Cell Signaling (Danvers, MA, USA), and ORIGENE (Rockville, MD, USA), respectively.

### 2.7. Statistical Analysis

Unpaired t-tests and Fisher’s exact tests were used to evaluate the differences between the two groups. The chi-square test was used to evaluate differences between three or more groups. Survival curve analysis was performed using the Kaplan–Meier method, and statistical significance was calculated using the log-rank test and the Gehan–Breslow–Wilcoxon test. All analyses were performed using the GraphPad Prism 9 software (GraphPad Software, San Diego, USA). Differences were considered significant at * *p* < 0.05, ** *p* < 0.01, *** *p* < 0.001.

## 3. Results

### 3.1. PDX Models Improved the Success Rate of Tumor Engraftment and Growth Rate with Increased In Vivo Passage

To establish PDX models of gynecologic cancer, we tested ovarian cancer (*n* = 130), cervical and vaginal cancer (*n* = 45), and uterine cancer (*n* = 32) (Table 1). In the transplantation of the patient tumor to mice, successful engraftment rates of ovarian cancer, cervical and vaginal, and uterine cancer showed 46.92%, 64.44%, and 56.25% success rates, respectively. It took 5–7 months for the growth of the tumors in the mice. As the next mice were passaged, the tumor engraftment rate increased, and the duration required for engraftment was shortened. Moreover, the tumor growth rate increased depending on PDX passage in ovarian and uterine cancers but did not change in cervical and vaginal cancers. Next, we confirmed that the established PDX models maintained the histology of the original tumor via H&E staining (Appendix A).

### 3.2. Tumor Grade has a Decisive Effect on the Engraftment of PDX Tissues

To identify factors affecting PDX engraftment, we analyzed engraftment (P1) and the growth rate of PDX (P1) according to the clinical information of gynecologic cancers. First, we confirmed that cancer antigen 125 (CA-125), a blood marker for ovarian cancer, was significantly elevated in patients with ovarian cancer with successful PDX but not in those with failed PDX (Table 2, *p* = 0.0216). Borderline tumors did not have a good engraftment success rate, but invasive epithelial ovarian cancer accounted for the most successful cases, with 54 of 112 (48.21%) cases being successful. The success rate of PDX in epithelial ovarian cancer is affected by the cell type, stage, and tumor grade. We confirmed that cases with a higher stage (*p* = 0.0004) and worse grade (*p* = 0.0073) had a much greater PDX success rate. In addition, the growth rate of PDX tumors was faster in the late stage than in the early stage (*p* = 0.0002). The number of non-epithelial ovarian cancers that attempted to establish PDX was small (*n* = 8), but PDX models were established for germ cells and metastatic cell types (Table 2). In cervical cancer, P1 engraftment showed a success rate of 62.79% (27 of 43 cases), and vaginal cancers were attempted in two cases, both of which were successful. The PDX success rate of cervical cancer and vaginal cancer was also higher with poorer tumor differentiation (*p* = 0.0073), but there was no difference according to cell type, stage, tumor growth rate, and squamous cell carcinoma antigen (SSC Ag) (Table 3). Finally, in uterine cancer, 18 of 32 cases were successful (56.25%), and the PDX success rate depended on tumor grade (*p* = 0.0025), but there was no correlation with other factors (Table 4). In conclusion, we found that the common clinical factor influencing the establishment of PDX was tumor grade, regardless of tumor histotype.

### 3.3. PDX Engraftment Rate and Tumor Growth Rates Correlate with Poor Prognosis

To confirm the correlation between the prognosis of patients and P1 tumor engraftment and growth rates, we evaluated the overall survival of patients according to tumor engraftment status. First, the overall survival of patients with ovarian cancer was worse in the successful PDX group than in the failed group (HR = 2.107, *p* = 0.0371) (Figure 1A). As a result of prognostic analysis of epithelial ovarian cancer, which accounts for 85~90% of ovarian cancer [34], it was found that the successful group had a poor prognosis. However, there was no difference within Stage III patients (Appendix A). The overall survival of patients with cervical cancer tended to be poorer in the successful PDX group than in the failed group. However, these results were not statistically significant (HR = 4.333, *p* = 0.0774) (Figure 1B). In contrast, there was no change in the overall survival of patients with uterine cancer (Figure 1C). Furthermore, we analyzed the overall survival according to the PDX tumor growth rate (P1). The prognosis was worse in patients with ovarian cancer with fast tumor growth than in those with slow tumor growth (Figure 2A). Notably, the same prognostic results were shown when only the epithelial ovarian cancer and Stage III cancers were targeted (Appendix A). In contrast, there was no association between patients’ survival and tumor tissue engraftment in cervical, vaginal, or uterine cancers (Figure 2B,C). These results suggest that the PDX success rate and tumor growth rate are associated with a poor prognosis in ovarian cancer.

### 3.4. Inflammation and Immune Response Genes Correlate with the PDX Establishment Rate

To identify genes that influenced the establishment of PDX, we analyzed the differentially expressed genes (DEGs) using RNA sequencing data from ovarian cancer tissues (GSE157153). GSE157153 is a patient tissues dataset of clear-cell-type ovarian cancer registered with the Gene Expression Omnibus, which we reported previously [35]. Among the cancer tissues of GSE157153, the tissues of four patients with clear cell ovarian cancer concurred with cancer tissues to establish PDXs (Table 5). Two patient tissues were successfully engrafted, but the others failed. To select DEGs between the tissues with PDX success and failure, we performed GSEA. As a result, we obtained 50 hallmark gene sets from GSEA, and 24 gene sets were correlated with the successful PDX group. Notably, the high-ranking gene sets were related to inflammation and immune responses (Figure 3A). We confirmed the list of upregulated genes of TNFA_SIGNALING_VIA_NFKB, including NFKB, which is the most important regulator of the inflammatory response, using a heatmap (Figure 3B). They showed higher mRNA expression in the successful tissues than in the PDX-failed tissues. To evaluate the protein levels in the tissues, we performed immunohistochemistry for three genes: NFKB2, RELB, and ICOSLG. This confirmed that the protein levels of NFKB2, RELB, and ICOSLG were higher in the success group than in the failure group (Figure 3C). Therefore, these results indicate that PDX engraftment is positively correlated with inflammation- and immune-response-related genes, and that the identified genes can influence the PDX engraftment rate.

### 3.5. Up-Regulated Genes in Successful Engraftment Are Associated with a Poor Prognosis of Patients with Ovarian Clear Cell Cancer

We extracted only ovarian clear cell cancer (*n* = 37) from gene expression profiling results (GSE73614) and carried out Kaplan–Meier analysis about up-regulated genes in successful engraftment of PDXs. The Kaplan–Meier curves indicated that the overall survival tends to be unfavorable in high-NFKB2-expression patients, but this was not validated by the statistics (Figure 4A), nor did the RELB expression affect overall survival (Figure 4B). Patients with high ICOSLG expression have a significantly poorer prognosis than those with low expression of ICOSLG (*p* = 0.0448) (Figure 4C). Our findings showed that the DEGs obtained according to PDX engraftment status could be potential targets to the prognosis marker of ovarian clear cell cancer patients.

## 4. Discussion

Patient-derived tumor xenograft (PDX) models retain certain characteristics of each patient, such as gene mutations, gene expression profiling, and pathological features [15]. In addition, they accurately replicate the results of corresponding therapies received by patients [36]. Therefore, PDX models have many roles in precision medicine, and many gynecologic cancer PDX models have also been established for preclinical or clinical research [29].

We attempted subcutaneous transplantation of more than 200 gynecologic cancer cases to BALB/c nude mice, suggesting a PDX success rate similar to the rate reported in previous studies [37]. Although few cases of borderline tumor and vaginal cancer were included, we established PDX models of these tumors, which have low malignancy potential and limited occurrence, respectively.

It is essential to use immunodeficient mice for the establishment of PDX models because the genetic background of mice is a crucial factor influencing PDX success [38]. Since nude mice were introduced in 1966 [39], immunodeficient mice have been continuously improved, including athymic nude, SCID, NOD/SCID, and NOG according to immune-defected grade by genetic modifications. Athymic nude mice have a mutation at the Foxn1 (winged-helix/forkhead transcription factor) gene, which blocks thymus-derived T cells, but have highly activated NK cells. SCID mice have a mutation in Prkdc/scid (protein kinase, DNA activated, catalytic polypeptide) protein and a lack of mature B and T lymphocytes. NOD/SCID mice show many innate immune defects, including NK cell dysfunction, low cytokine production, and T and B cell dysregulation [40]. NOG mice were developed most recently. Human cell and tissue transplantation are more commonly performed using NOG mice than other mouse models because NOG mice have defects in T cells, B cells, and NK cells [41]. However, NOG mice greatly increase the cost of the PDX model study. The success rates of the PDX model in gynecologic cancer in NOG showed a 46.7% success rate [37]. PDX success rates in SCID mice are 74% for ovarian cancer and 48–70% for cervical cancer [22,23,24]. Our results confirmed a 52% success rate of PDX models in BALB/c nude mice, suggesting that PDX models of gynecologic cancer did not correspond with the success rate of cancer tissue transplantation in previously reported immunodeficient grades. Therefore, establishing PDX using BALB/c nude mice in this study might reduce costs and increase effectiveness.

We obtained the hallmark gene sets for gene alterations between successful and failed PDX tissues using GSEA. Upregulated gene sets in successful PDX tissue showed the signal pathways of “immune and inflammation response: “allograft_rejection”, “TNFA_signailing_via_NFKB”, “interferon_GAMMA_response”, “interferon_alpha_response”, “inflammatory_response”, and “IL2_STAT5_SIGNALING”. These results suggest that PDX engraftment status distinguishes the gene expression pattern associated with patients’ immunity and inflammation.

The results of GSEA, the nuclear factor kappa-light-chain-enhancer of activated B cell (NF-KB) pathways within an upregulated gene set, could be correlated with tissue engraftment and growth. NF-κB signaling is activated by tumor necrosis (TNF), IL-1, and toll-like receptor ligands. NF-κB is a protein complex that controls the gene transcription involved in “cell proliferation and survival,” “epithelial to mesenchymal transition,” invasion, angiogenesis, metastasis, and inflammation-related genes [42,43]. Additionally, NF-κB regulates cell adhesion molecules that promote tumor growth and metastasis in cancer cells [44]. The high expression of adhesion molecules E-selectin, VCAM-1, and ICAM-1 by NF-κB activation assists the attachment of white blood cells to the stimulated endothelial cell surface [45,46]. According to a report, NF-κB signaling appears to be highly activated in cancer-associated fibroblasts (CAFs), consisting of tumor microenvironments. Here, the tumor growth delay by inhibiting NF-kB activation was examined in a xenograft model [47,48]. PDX models can reenact the tumor microenvironment; in these models, activation of NF-κB is considered to have an important role in tissue growth. The upregulation of adhesion molecules by activating NF-κB could have positive effects on the engraftment of patient tissue to mouse subcutaneous tissues in the early stages of tissue transplantation.

We confirmed that PDX engraftment rate and tumor growth rates correlate with poor prognosis in ovarian cancer. Based on this, we considered that DEGs with high expression in the tumor of ovarian clear cell patients engrafted successfully could be potential prognosis markers. ICOSLG predicted overall survival with ovarian clear cell cancer via the Kaplan–Meier curve in our study. Unfortunately, our results did not check the role of NFKB2 and RELB, but previous reports showed they could be candidates as prognosis markers. NFKB2 is reportedly associated with poor ovarian cancer [49]. Moreover, the loss of RELB in mouse xenograft models using ovarian cancer cell lines significantly decreased chemoresistance, tumorigenesis, and ALDH expression and activity [50,51]. We suggest that a new prognosis marker can develop from the DEGs obtained according to PDX engraftment status.

There are two limitations to the interpretation of the results of this study. First, we performed our experiments for five years to establish the PDX models, but we did not statistically validate the overall survival from cervical and vaginal, and uterine cancer due to the small number of patients. Second, GESA analysis was conducted in only four cases with clear cell ovarian cancer in this study. To overcome these limitations, we will continue establishing PDX models and further collect gene profiling from various subtypes of cancer. Accordingly, we expect to obtain significant data from all gynecologic cancer types with ovarian cancer.

## 5. Conclusions

Our study concluded that immune- and inflammation-related genes are strongly associated with ovarian cancer PDX engraftment rate. In addition, they can predict the survival of patents with ovarian cancer. Further studies are needed to characterize patients’ cancer tissues, including corresponding PDX cancer tissues, utilizing sequencing technology. The gynecologic cancer PDX tissues that we established might be crucial to validate and evaluate anti-cancer drug responses in patients who develop cancer in the future.

## Figures and Tables

**Figure 1 cancers-14-00829-f001:**
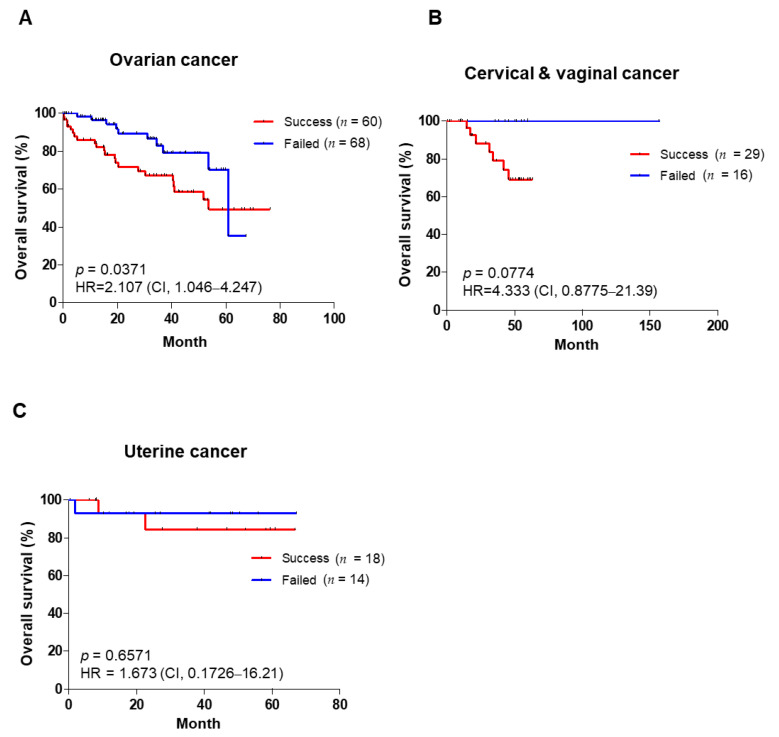
Overall survival of patients based on tumor engraftment status. Overall survival of patients with ovarian cancer (**A**), cervical and vaginal cancer (**B**), and uterine cancer (**C**) in the success groups and failed groups of P1. Kaplan–Meier survival analysis was carried out, and log–rank *p*–values, HR, and CI are shown for each of the results. HR: hazard ratio; CI: confidence interval.

**Figure 2 cancers-14-00829-f002:**
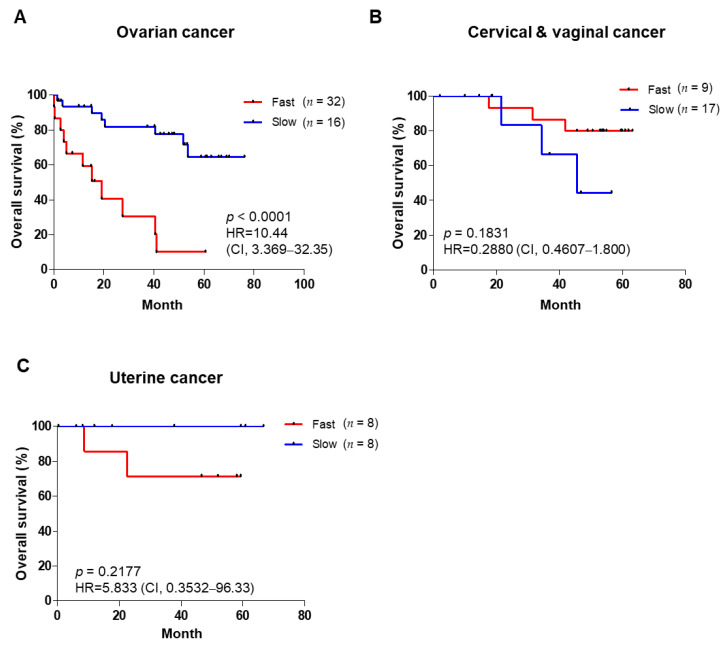
Overall survival of patients based on tumor growth rate. Overall survival of patients with ovarian cancer (**A**), cervical and vaginal cancer (**B**), and uterine cancer (**C**) in the slow and fast groups of P1 tumors. Kaplan–Meier survival analysis was carried out, log–rank *p*–values, HR, and CI are shown for each of the results. HR: hazard ratio; CI: confidence interval.

**Figure 3 cancers-14-00829-f003:**
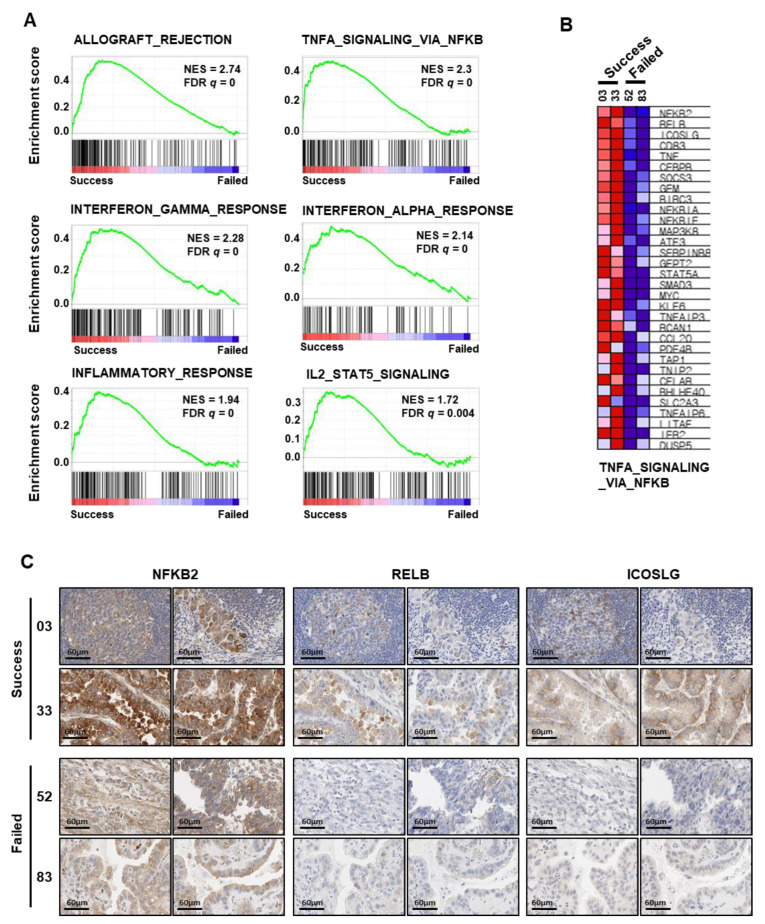
Analysis of DEGs between successful and failed tumor engraftment in patient tissues. (**A**) GSEA of RNA-sequencing data for clear cell ovarian cancers with successful engraftment (*n* = 2) versus failed engraftment (*n* = 2) from the GSE157153 dataset. (**B**) Heatmap presents up-regulated DEG lists in TNFA_SIGNALING_VIA_NFKB gene set. (**C**) Immunohistochemistry of NFKB2, RELB, and ICOSLG in tissues of patients with clear cell ovarian cancer with successful engraftment or failed engraftment. Immunohistochemistry images show two parts of each tissue (×400). DEG: differentially expressed genes; GSEA: gene set enrichment analysis.

**Figure 4 cancers-14-00829-f004:**
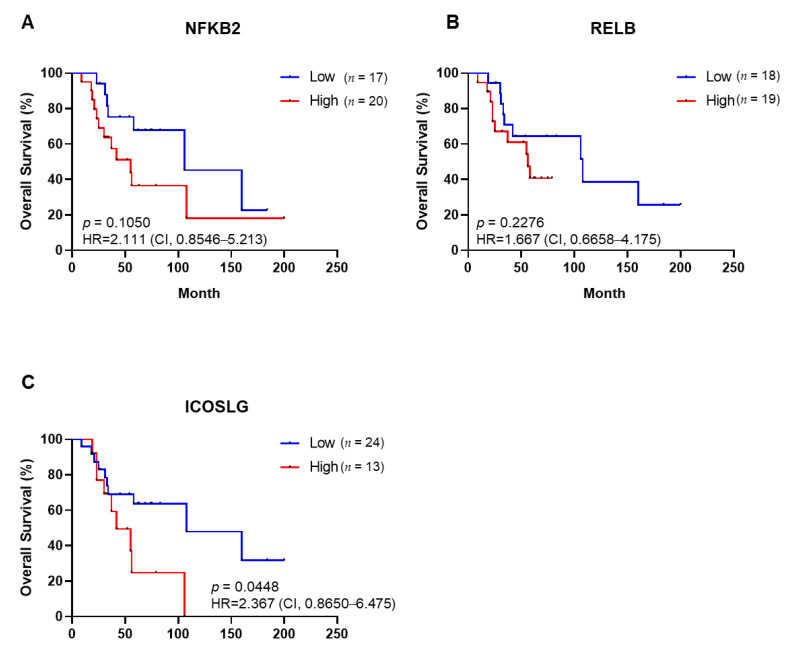
Overall survival of ovarian clear cell cancer patients based on gene expression levels of NFKB2 (**A**), RELB (**B**), and ICOSLG (**C**), respectively. Kaplan–Meier survival analysis was carried out; log–rank *p*–values, HR, and CI are shown for each of the results. HR: hazard ratio; CI: confidence interval.

**Table 1 cancers-14-00829-t001:** Comparison of tumor engraftment and tumor growth rate according to the development of patient-derived xenograft models.

						Successful Engraftment			
Diagnosis	Case	Successful Engraftment	Failed Engraftment	Successful Engraftment Rate (%)	*p* Value ^a^	Duration ^b^ (Month)	*p* Value ^c^	Tumor Growth Rate ^b^	*p* Value ^d^
Ovarian cancer, *n*					<0.0001		0.0021		
P1	130	61	69	46.92		6.70 ± 3.95		319.3 ± 319.7	reference
P2	61	55	6	90.16		4.93 ± 3.55		575.1 ± 613.3	0.0145
P3	55	52	3	94.55		4.09 ± 2.85		527.5 ± 457.9	0.0168
Cervical and Vaginal cancer, *n*					0.0004		0.0007		
P1	45	29	16	64.44		5.42 ± 2.74		606.7 ± 527.0	reference
P2	29	26	3	89.66		3.19 ± 1.57		558.7 ± 594.6	0.7828
P3	26	26	0	100		3.10 ± 1.61		525.2 ± 394.5	0.5891
Uterine cancer, *n*					0.014		0.08		
P1	32	18	14	56.25		6.04 ± 4.13		427.3 ± 507.2	reference
P2	18	17	1	94.44		4.11 ± 3.15		449.8 ± 339.0	0.8992
P3	17	13	4	76.47		2.80 ± 1.48		1208 ± 1204	0.0341

^a^ Chi-square test values for success and failure. ^b^ Values are mean ± standard derivation. ^c^ One-way ANOVA test among P1, P2, and P3. ^d^ Unpaired t-test values for growth rate of P2 or P3 compared to P1.

**Table 2 cancers-14-00829-t002:** Clinicopathological characteristics of patients with ovarian cancer according to P1 tumor engraftment status.

					Successful Engraftment	
Variables	Case	Successful Engraftment	Failed Engraftment	*p* Value	Tumor Growth Rate ^a^	*p* Value ^d^
All, *n*	130	61	69			
Age (year) ^a^	52.12 ± 12.22	52.08 ± 12.51	52.16 ± 12.05	0.9706 ^b^		
Pre CA-125 level (U/mL) ^a^	942.7 ± 2315	1442 ± 3124	475.4 ± 942.7	0.0216 ^b^		
Borderline tumors, *n*	10	2	8			
Histology, *n* (%)				1.0000 ^c^		
Serous	2 (20.00)	0 (0.0)	2 (25.00)		N/A	N/A
Mucinous	8 (80.00)	2 (100)	6 (75.00)		167.55 ± 102.2	
Malignant tumors, *n*	120	59	61			
Epithelial ovarian cancer, *n*	112	54	58			
Histology, *n* (%)				0.0389 ^c^		
Serous	68 (60.71)	39 (72.22)	29 (50.00)		300.9 ± 248.3	reference
Clear	17 (15.18)	6 (11.11)	11 (18.97)		395.4 ± 383.1	0.6088
Endometrioid	7 (6.25)	2 (3.70)	5 (8.62)		90.28 ± 2.653	<0.0001
Mucinous	12 (10.71)	2 (3.70)	10 (17.24)		373.5 ± 322.0	0.807
MMMT	7 (6.25)	5 (9.26)	2 (3.45)		632.7 ± 696.4	0.059
Brenner	1 (0.89)	0 (0.0)	1 (1.72)		N/A	N/A
Stage, *n* (%)				0.0004 ^c^		
I	27 (24.11)	5 (9.26)	22 (37.93)		112.6 ± 53.89	reference
II/III/IV/Recurrent	85 (75.89)	49 (90.74)	36 (62.07)		359.3 ± 335.1	0.0002
Grade, *n* (%)				0.0073 ^c^		
1	8 (7.14)	0 (0.0)	8 (13.79)		N/A	
2	10 (8.93)	7 (12.96)	3 (5.17)		316.6 ± 166.8	reference
3	70 (62.50)	37 (68.52)	33 (56.90)		332.5 ± 287.1	0.2044
Unknown	24 (21.43)	10 (18.52)	14 (24.14)		443.7 ± 521.4	N/A
Non-Epithelial ovarian cancer, *n*	8	5	3			
Histology, *n* (%)				0.2369 ^c^		
Sex cord-stromal	1 (12.50)	0 (0.0)	1 (33.33)		N/A	N/A
Germ cell	5 (62.50)	3 (60.00)	2 (66.67)		84.12 ± 106.3	
Metastatic	2 (25.00)	2 (40.00)	0 (0.0)		N/A	
Stage, *n* (%)				0.1964 ^c^		
I	5 (62.50)	2 (40.00)	3 (10.00)		159.25	N/A
II/III/IV/Recurrent	3 (37.50)	3 (60.00)	0 (0.0)		8.98	

^a^ Values are mean ± standard derivation. ^b^ Unpaired t-test for success and failure. ^c^ Chi-square test or Fisher’s exact test values for success and failure. ^d^ Unpaired t-test with Welch’s correction. Unknown data was excluded in calculation of *p* value. N/A, not applicable.

**Table 3 cancers-14-00829-t003:** Clinicopathological characteristics of patients with cervical or vaginal cancer according to P1 tumor engraftment status.

					Successful Engraftment	
Variables	Case	Successful Engraftment	Failed Engraftment	*p* Value	Tumor Growth Rate ^a^	*p* Value ^e^
All, *n*	45	29	16			
Age (year) ^a^	48.86 ± 11.70	49.04 ± 10.64	48.56 ± 13.73	0.8991 ^c^		
SCC Ag (ng/mL) ^a^	15.05 ± 33.27	13.74 ± 18.15	17.68 ± 52.87	0.7225 ^c^		
Diagnosis, *n* (%)				0.5313 ^d^		
Cervical cancer	43 (95.56)	27 (93.10)	16 (100)		603.9 ± 548	reference
Vaginal cancer	2 (4.45)	2 (6.90)	0 (0.0)		639.8 ± 183	0.8532
Histology, *n* (%)				0.2238 ^d^		
Squamous	32 (75.86)	22 (75.86)	10 (62.5)		667.7 ± 570.9	reference
Adeno	9 (20.00)	6 (20.69)	3 (18.75)		399.0 ± 326.8	0.1936
Others ^b^	4 (8.89)	1 (3.45)	3 (18.75)		424.4	N/A
Stage, *n* (%)				0.7551 ^d^		
I/II	20 (44.45)	12 (41.38)	8 (50.00)		591.6 ± 495.3	reference
III/IV/Recurrent	20 (55.00)	17 (58.62)	8 (50.00)		617.8 ± 566.0	0.9013
Grade, *n* (%)				0.0073 ^d^		
1	3 (6.67)	0 (0.0)	3 (18.75)		N/A	
2	14 (31.11)	10 (34.48)	4 (25.00)		873.2 ± 588.3	reference
3	4 (8.89)	4 (13.79)	0 (0.0)		925.5 ± 772.0	0.9237
Unknown	24 (53.33)	15 (51.72)	9 (56.25)		328.1 ± 225.7	N/A

^a^ Values are mean ± standard derivation. ^b^ Others include two cases of adenosquamous carcinoma, two cases of mucinous. ^c^ Unpaired t-test for success and failure. ^d^ Fisher’s exact test or chi-square test. ^e^ Unpaired t-test with Welch’s correction. Unknown data was excluded in calculation of *p* value. N/A, not applicable.

**Table 4 cancers-14-00829-t004:** Clinicopathological characteristics of patients with uterine cancer according to P1 tumor engraftment status.

					Successful Engraftment	
Variables	Case	Successful Engraftment	Failed Engraftment	*p* Value	Tumor Growth Rate ^a^	*p* Value ^e^
All, *n*	32	18	14			
Age (year) ^a^	56.44 ± 10.01	58.44 ± 8.361	53.86 ± 11.60	0.2032 ^c^		
Pre CA-125 level (U/mL) ^a^	477.7 ± 1574	307.7 ± 914.9	740.3 ± 2286	0.4880 ^c^		
Histology, *n* (%)				0.1595 ^d^		
Endometrioid	22 (68.75)	10 (55.56)	12 (85.71)		288.4 ± 342.9	reference
Serous	5 (15.63)	4 (22.22)	1 (7.14)		429.5 ± 532.7	0.7091
Clear	1 (3.13)	1 (5.56)	0 (0.0)		338.2	N/A
Carcinosarcoma	3 (9.38)	3 (16.67)	0 (0.0)		1127 ± 1088	0.476
Other ^b^	1 (3.13)	0 (0.0)	1 (7.14)		N/A	N/A
Stage, *n* (%)				1.0000 ^d^		
I/II	23 (71.88)	13 (72.22)	10 (71.43)		363.9 ± 559.0	reference
III/IV/Recurrent	9 (28.13)	5 (27.78)	4 (28.57)		617.8 ± 275.4	0.2565
Grade, *n* (%)				0.0025 ^d^		
1	4 (12.50)	0 (0)	4 (12.50)		N/A	
2	13 (40.63)	5 (15.63)	8 (25.00)		330.1 ± 477.3	reference
3	11 (34.38)	10 (31.25)	1 (3.16)		358.4 ± 293.8	0.919
Unknown	4 (12.50)	3 (9.38)	1 (3.16)		966.4 ± 1316	N/A

^a^ Values are mean ± standard derivation. ^b^ Other includes endometrial stromal sarcoma. ^c^ Unpaired t-test for success and failure. ^d^ Fisher’s exact test or chi-square test. ^e^ Unpaired t-test with Welch’s correction. Unknown data was excluded in calculation of *p* value. N/A, not applicable.

**Table 5 cancers-14-00829-t005:** Four patients belonging to both PDX and GSE157153.

Patient NO.	Diagnosis	Histology	Age	Stage	GSE157153	P1 Tumor Engraftment
03	ovarian cancer	clear cell	47	R	GSM4756750	Success
33	ovarian cancer	clear cell	32	IC	GSM4756737	Success
52	ovarian cancer	clear cell	41	IV	GSM4756755	Failed
83	ovarian cancer	clear cell	44	IA	GSM4756757	Failed

R; recurrent.

## Data Availability

The data presented in this study are available on request from the corresponding author.

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
