# Peer review of "Identification of Prognostic Markers of Gynecologic Cancers Utilizing Patient-Derived Xenograft Mouse Models"

_cancers, 2022, doi:10.3390/cancers14030829_

Round 1
Reviewer 1 Report
The manuscript titled "Identification of prognostic markers of gynecologic cancers utilizing patient-derived xenograft mouse models" is well designed and explained very details of the experiment. The results will be more interesting to the readers about the gynecology PDX engraftment success rate, the role of TNF pathways ( NFKB2 and RELB) behind the success of Ovarian cancers engraftment rate. However, I have some minor concerns that need to be addressed before recommending them for publication.
1) Is there any previous studies in gynecology cancers that observed a similar trend of growth rate increase for Uterine and Ovarian cancer PDX models?
2) Line 233-238: authors mentioned no association found between cervical/vaginal/uterine cancers in patient's survival and tissue engraftment except Ovarian cancer. Line 238 was written as gynecologic cancer, should be changed to " Ovarian cancer" as data only supports that.
Reviewer 2 Report
The manuscript by Shin H describes a xenobak of patient derived xenografs (PDXs) of gynaecological cancers. The authors established about one hundred of PDX and state that immune and inflammation related genes were strongly correcaleted with both ovarian cancer engraftment and could predict survival of cancer patiente.
The manuscript aim was the establishment of various gynaecological PDX models and which were the factors influencing tumor take and tumor growth. There are however, major concerns that preclude the manuscript to be published in Cancers.
The authors described the establishment of PDX coming from different gynaecological cancers. I have big concerns in having all the gynaecological cancers gathered as each tumor type represents a different entity and should be considered as a single group. I would delete in table 1 the pooled analysis.
No data on the histology of the PDXs that were obtained: did they reflect the histology of the human tumor they derive from? Please add and comment the data. In addition no information on the lag period of tumor appearance. How long did it take for the tumors to growth? Was there any difference in these values at F1, F2 and F3?
In their cases of malignant tumors, there was a quite high number of mucinous ovarian carcinoma, whose incidence is generally 2-3%. Which were the criteria for the pathological diagnosis? I found difficult to interpret the data of Figure B that the overall survival data. Indeed the overall survival refers to all stages and all hystologies? Indeed it is known that tumors stage I have a good prognosis as regards to stages II/III/IV and the analysis should be re-done taking it into consideration. What about tumor residual tumor at surgery? Was this considered for tumor engraftment? The residual tumors is one of the most important clinical and this was the only parameter correlating with tumor engraftment in a paper not cited by the authors (Ricci F. et al, Cancer Res. 2014 Dec 1; 74(23):6980-90.), that also report the establishment of PDXs using nude mice.
As a general note, the survival curve based on tumor engraftment/and tumor growth reflect very few patients and I have some concerns of the statistical analysis done.
How was defined a fast tumor growth versus a slow tumor growth? No clear to me is the D tumor volume values. It is state both that the D values: were obtained dividing the tumor volume measured at the end of the tumor growth period and also that refers to tumor growth during one months (mm3/month). It is quite confusing is a ration or a difference of tumor volume).
I also found quite difficult to understand the reasoning leading to the identification of inflammation and immune response gene affecting PDX establishment. Indeed, the authors only focus on clear cell that do represent a specific subset of ovarian carcinoma and they consider stage I and relapsing, really very different clinical conditions. Why not focusing on high grade serous ovarian carcinomas, that are the most frequent and deadly ovarian carcinoma?
As a general not, the manuscript is very descriptive. While prognostic factor could be important, I personally believe that a PDX xenobank could have the chance to identify predictive value of response to therapy and explore new therapeutic strategies.
Reviewer 3 Report
Shin et al. use a large number of mice to establish xenografts from patient tissues (PDX) of various gynecologic cancers. Among the gynecologic cancers tested, the authors found that only ovarian cancer showed correlation of PDX graft success and pace of growth with survival of patients of the source tissue. While the authors show various other kinds of correlation, the true value and the most convincing aspect of this study is that the PDX models enabled the discovery of gene expression signatures that accurately predicted clinical prognosis of ovarian cancer cases. Essentially, projected to a clinical scenario, the prognosis of an ovarian cancer patient that gets admitted will be predicted by a quick gene expression profiling of these prognostic markers informed by PDX from this study. While that alone merits publication, the writing must be edited to not overstate some conclusions, state some things directly and give more information on interpreting some data.
Comments to address:
Lines 40-45: Unless the incidence of gynecologic cancer is specifically high in Korea, it is good to mention world statistics instead. Also, instead of stating that 25.4% of them died, please state the five-year survival rate or mention over what time period the deaths were recorded.
Lines 56-57: The writing can be more concise and direct. For example: “Furthermore, patients can increase the severity of disease by negligence or misjudgment of themselves” – this is both longwinded as well as vague. What do the authors mean by “negligence or misjudgement”?
Line 88: Rewrite the sentence to clarify what you mean by “apply preserved PDX tissues to future patients”…
Line 96: The premise is unclear. How would PDX give more information about the cancer than form the cancer biopsies themselves? The study does have its own value in that it is a comprehensive study of PDX models of gynecologic cancers. The authors must acknowledge that and not overstate the importance of the study.
Lines 125-128 and elsewhere on the nomenclature of tumor engraftment: Unless this is the standard way, I suggest a more intuitive way of calling them – for instance, primary, secondary, and tertiary engraftment instead of F1/F2/F3 because the mice are not generationally related.
Section 3.3:
The specific trends of each cancer type on prognosis vs PDX success/failure or PDX F1 growth rate have their own value. The authors should remove generalized conclusions on gynecologic cancers and PDX, as it is misleading and of less value than their specific characterization.
Figure 3A: Please explain how to interpret this panel.
Figure 3B is very nice. If possible, please include a similar analysis of gene expression signature of tumors that gave rise to fast versus slow growing PDX as well. This will be of immense value in understanding clinical prognosis.
Figure 3C: The value of this analysis is unclear. IHC on a failed graft is not probing live tumor tissue, right? My impression is that this is not a valid and informative comparison because of this reason.
Discussion, lines 374-384: Any talk of PDX models being useful for predictive drug efficacy is beyond the scope of this study and the authors must remove such discussion, to make the manuscript convey its importance clearly.
